# Impact of the Power-Dependent Beam Diameter during Electron Beam Additive Manufacturing: A Case Study with γ-TiAl

Marcel Reith [1,*] , Christoph Breuning [2] , Martin Franke [1] and Carolin Körner [2]

1   Neue Materialien Fuerth, Dr-Mack-Str. 81, 90762 Fuerth, Germany
2   Chair of Materials Science and Engineering for Metals, Friedrich-Alexander-Universität Erlangen-Nürnberg, Martensstr. 5, 91058 Erlangen, Germany
*   Correspondence: marcel.reith@fau.de

**Abstract:** The development of process parameters for electron beam powder bed fusion (PBF-EB) is usually made with simple geometries and uniform scan lengths. The transfer to complex parts with various scan lengths can be achieved by adapting beam parameters such as beam power and scan speed. Under ideal conditions, this adaption results in a constant energy input into the powder bed despite of the local scan length. However, numerous PBF-EB machines show deviations from the ideal situation because the beam diameter is subject to significant changes if the beam power is changed. This study aims to demonstrate typical scaling issues when applying process parameters to scan lengths up to 45 mm using a fourth generation γ-TiAl alloy. Line energy, area energy, return time, and lateral velocity are kept constant during the additive manufacturing process by adjusting beam power and beam velocity to various scan lengths. Samples produced in this way are examined by light microscopy regarding lateral melt pool extension, melt pool depth, porosity, and microstructure. The process-induced aluminum evaporation is measured by electron probe microanalysis. The experiments reveal undesired changes in melt pool geometry, gas porosity, and aluminum evaporation by increasing the beam power. In detail, beam widening is identified as the reason for the change in melt pool dimensions and microstructure. This finding is supported by numerical calculations from a semi-analytic heat conduction model. This study demonstrates that in-depth knowledge of the electron beam diameter is required to thoroughly control the PBF-EB process, especially when scaling process parameters from simply shaped geometries to complex parts with various scan lengths.

**Keywords:** electron beam powder bed fusion; electron beam; beam diameter; additive manufacturing; titanium aluminide; microstructure; evaporation

## 1. Introduction

The electron beam powder bed fusion (PBF-EB) is a well-established method in additive manufacturing (AM) [1–3]. The PBF-EB process can be divided into four steps: lowering the build platform, powder application, preheating the surface with a defocused electron beam, and melting with a focused electron beam. A detailed discussion of the PBF-EB process can be found in [1,3]. One of the great benefits of AM technology is a higher degree of design freedom compared to conventional manufacturing technologies [1,3] Hence, the final goal of most AM processes is the production of complex parts.

To manufacture a complex part, suitable process parameters have to be identified. Therefore, the scanning parameters beam power, beam velocity, and line offset are tested, mostly on cuboids with a constant scan length. Depending on the chosen parameters, a trailing or a persistent melt pool can form. The melt pool is classified as trailing if the melt pool is already solidified when the electron beam reaches the same position as the subsequent melt line. Contrarily, the melt pool is called persistent when the melt pool is still liquid. For a more detailed description, the reader is referred to [4]. Subsequently, the

samples are examined for defects and swelling and a processing window is established, as illustrated for Ti-6Al-4V by [5–7]. More detailed investigations revealed, that the choice of the scanning parameter can manipulate the microstructure within the processing window. For instance, the microstructure of Ni-based superalloys can be adjusted to equiaxed, columnar, or single-crystal by controlling the melt pool geometry and dimensions via different scanning strategies [8–11]. A further example is the aluminum evaporation in titanium aluminides during the PBF-EB process, also controlled by the melt pool [12,13]. In conclusion, the melt pool is the determining factor for part properties such as defects and microstructure.

Consequently, a uniform melt pool with constant solidification conditions at every position is necessary to produce a homogeneous microstructure in a complex part. A common approach to achieving a constant melt pool is using constant thermal conditions and constant energy input for melting. Firstly, to ensure constant thermal conditions, the beam velocity $v_B$ is adjusted to the scan length $l_{sthat}$ so that the mean return time $t_r$ is constant (see Equation (1)). The mean return time $t_r$ is defined as the average time the electron beam needs to come back to adjacent points of two subsequent scan lines. This ensures that the temperature field during melting remains on average constant, independently of the scan length for a cross-snake hatch pattern [4]. It is to note that for a constant line offset $l_o$, the lateral velocity $v_{lat}$, which is the velocity of the melt pool perpendicular to the electron beam scanning direction (see Figure 1c), stays unchanged if the return time $t_r$ is also constant (Equation (2)).

$$t_r = \frac{l_s}{v_B} \tag{1}$$

$$v_{lat} = \frac{l_o}{t_r} = \frac{v_B * l_o}{l_s} \tag{2}$$

As a measure for the energy input the line energy $E_L$ and the area energy $E_A$ are introduced (see Equations (3) and (4)). To account for the increasing and decreasing beam velocity $v_B$ due to longer and shorter scan length $l_s$, the beam current $I_B$ and, respectively, the beam power $P$ is adjusted to keep the line energy $E_L$ constant. Since there is no PBF-EB system available which can adjust the acceleration voltage $U_B$ during the process, this is the only way to keep the line energy $E_L$ steady. Moreover, if the line offset $l_o$ is fixed, the line energy $E_L$ and the area energy $E_A$ are proportional, and hence the area energy $E_A$ is kept constant as well.

$$E_L = \frac{U_B * I_B}{v_B} = \frac{P}{v_B} \tag{3}$$

$$E_A = \frac{E_L}{l_o} = \frac{U_B * I_B}{v_B * l_o} \tag{4}$$

This scaling approach was successfully used for a fourth generation $\gamma$-TiAl alloy in the author's previous work. Cylinders with scan lengths of roughly 3–17 mm were manufactured without misconnections and displayed a homogenous microstructure and isotropic mechanical properties [14]. Similar findings have been reported in a numerical study conducted with a semi-empirical heat conduction model on Ti-6Al-4V by Breuning et al. [4]. Based on his results, the melt pool depth and width remain the same over a large range of scan lengths if the line energy and lateral velocity are kept constant [4]. In contrast, experimental studies showed that the scaling of process parameters can cause defects and influence the microstructure [15,16].

An important process parameter is the beam diameter. Although this is known in the PBF-EB community, very little has been published on this topic. One rare exception is the formulation for beam widening on an *Arcam AB* PBF-EB system suggested by Klassen et al. (Equation (5)) [12].

$$d = 4\sigma = \begin{cases} 342.5 \ \mu\text{m} + 0.11 \frac{\mu\text{m}}{W} * P, & P \leq 675 \ W \\ 20.6 \ \mu\text{m} + 0.58 \frac{\mu\text{m}}{W} * P, & P > 675 \ W \end{cases} \tag{5}$$

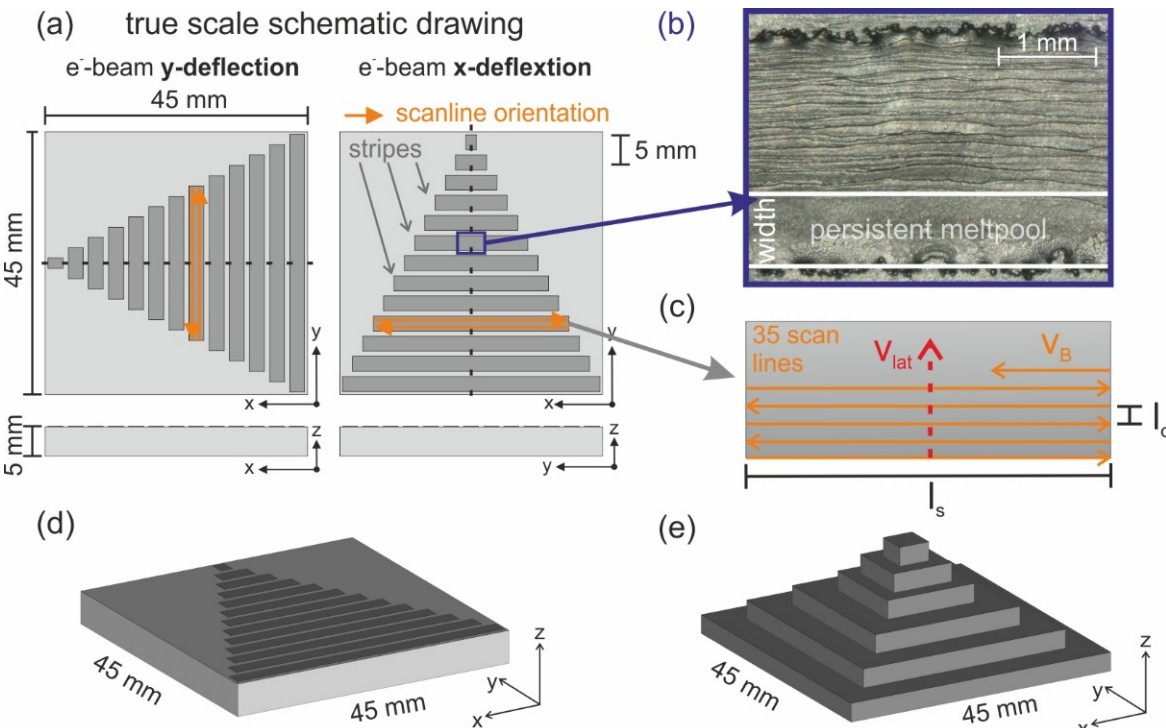

**Figure 1.** (**a**) Scaled schematic of the samples for determining the melt pool width and depth. Thirteen stripes (dark grey) consisting of 35 scan lines (orientation indicated by orange arrows) per examined beam power are molten for the x- and y-deflection of the electron beam. The single, 100 μm layer is molten on top of a 45 mm × 45 mm × 5 mm base plate (light grey) to reach common PBF-EB process conditions as close as possible. The section plane is indicated by the dotted lines (black). (**b**) Exemplary image of the persistent melt pool taken in the middle of the stripe. The melt pool width is defined as depicted. (**c**) Snake scan strategy used to build the samples. The beam velocity $v_B$, line offset $l_o$, scan length $l_s$, and lateral velocity $v_{lat}$ are indicated. (**d**) 3D CAD model of the geometry for melt pool investigations. (**e**) 3D CAD model of the geometry used for analyzing the bulk properties. The square plates possess an edge length between 45 and 2.5 mm.

Based on these values the beam diameter $d$, defined as $4\sigma$ of a Gaussian distribution, increases from 350 μm at 60 W beam power to 845 μm at 1500 W beam power (Figure 2). This means that the energy of the electron beam is distributed over a strongly increased surface for higher beam powers. However, the energy density is not only dependent on the beam diameter and the Gaussian distribution but also the power of the electron beam and hence is strongly influenced by the method of scaling the scanning parameters. To summarize, a wider electron beam is expected to influence the energy density and energy distribution of the electron beam and potentially can influence the melt pool.

The objective of this study is to improve the general understanding of how the beam diameter, which changes with beam power, influences PBF-EB. Therefore, not only the bulk properties such as misconnections, gas porosity, and aluminum content are examined but also the width and depth of the melt pool. By scaling previously determined scanning parameters [14], the energy input is kept constant while the beam power and beam velocity are adapted to the various scan length. Consequently, the beam diameter is the only remaining parameter, which can influence the resulting part properties. For the first time, the connection between beam diameter, power density, melt pool, and microstructure is systematically analyzed for PBF-EB. The influence of the beam diameter is not limited to the machine and material presented here, but has to be taken into account for every PBF-EB process on any machine. In the future, on-demand microstructure via electron-beam powder bed fusion will enable tailoring the mechanical properties of certain areas in complex parts towards their specific requirements [17]. Hence, this novel approach

will become even more important to ensure the needed microstructures are stable and homogenous.

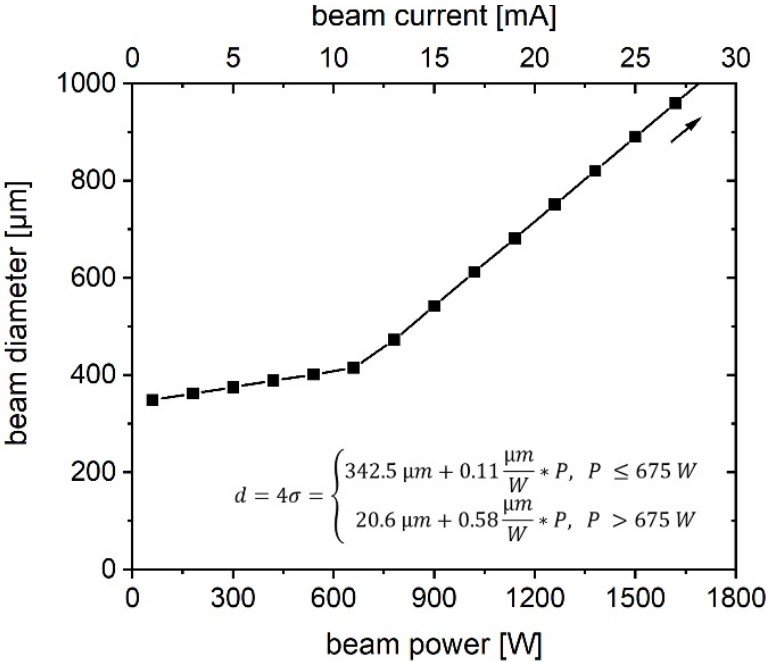

**Figure 2.** Beam diameter over beam power is calculated by Equation (5) according to [12].

## 2. Materials and Methods

Experiments are performed with two powder batches of the same γ-TiAl alloy, which only differ slightly in aluminum content (Table 1). The primary rods are produced by GfE Metalle und Materialien GmbH (Nuremberg, Germany). The rods are gas atomized into spherical powder with a particle size distribution from 50 μm to 120 μm by TLS Technik GmbH & Co. Spezialpulver KG (Bitterfeld, Germany).

**Table 1.** Chemical composition of the used powder batches While the nominal (nom.) composition is equal, the measured (meas.) composition differs slightly.

| γ-TiAl BMBF3 [1] | | Ti at.% | Al at.% | Nb at.% | W at.% |
|---|---|---|---|---|---|
| batch 1 | nom. | bal. | 47.5 | 5.5 | 0.5 |
| batch 1 | measured | bal. | 47.5 | 5.6 | 0.4 |
| batch 2 | nom. | bal. | 47.5 | 5.7 | 0.5 |
| batch 2 | measured | bal. | 46.8 | 5.7 | 0.5 |

[1] The alloy was developed during the ProMat_3D: NextTiAl project 03XP0088 founded by the Feder Ministry of Education and Research of Germany.

The powder is processed on an Arcam A2X (Arcam AB, Mölndal, Sweden) with the software version EBM Control 5.2. The x-deflection of the electron beam is parallel to the direction of rake movement, while the y-deflection is perpendicular to that. Thirteen defined stripes, each consisting of 35 scan lines with a line offset $l_o$ of 75 μm, are molten in a single, 100 μm thick layer, see Figure 1a. The single layer with the stripes is molten on top of a 5 mm thick plate with an edge length of 45 mm to get as close to the melting conditions of real parts as possible. To investigate the melt pool during hatching, 35 scan lines are chosen to get into a steady melting state in contrast to measuring single scan lines. Each stripe has beam power between 60–1500 W with a step size of 120 W. The scan lengths inside each stripe are set in a way to provide integer beam power. The beam velocity is adjusted to the beam power ensuring constant line energy, area energy, return time, and

lateral velocity (see Table 2). The process parameter set used as a basis was evaluated in preliminary experiments for scan lengths from roughly 3–7 mm and produced dense samples without misconnections, a homogeneous microstructure, and isotropic mechanical properties [14].

**Table 2.** Process parameters for the electron beam melting. The line offset 75 μm, acceleration voltage 60 kV, line energy 0.13 J/mm, area energy 1.70 J/mm$^2$, return time 3.75 ms, and lateral velocity 20 mm/s are constant by choosing specific scan lengths, which enable only using integer beam currents from 1 mA to 25 mA with a step size of 2 mA while adjusting the beam velocity.

| Scan Length | Beam Current | Beam Power | Beam Velocity |
|---|---|---|---|
| mm | mA | W | mm/s |
| 1.8 | 1.0 | 60 | 479 |
| 5.3 | 3.0 | 180 | 1412 |
| 8.8 | 5.0 | 300 | 2344 |
| 12.3 | 7.0 | 420 | 3276 |
| 15.9 | 9.0 | 540 | 4235 |
| 19.4 | 11.0 | 660 | 5167 |
| 22.9 | 13.0 | 780 | 6099 |
| 26.5 | 15.0 | 900 | 7058 |
| 30.0 | 17.0 | 1020 | 7990 |
| 33.5 | 19.0 | 1140 | 8922 |
| 37.1 | 21.0 | 1260 | 9881 |
| 40.6 | 23.0 | 1380 | 10,813 |
| 44.1 | 25.0 | 1500 | 11,745 |

The melt pool width is determined in the middle of the as-built samples with a light microscope Zeiss Axioskop 2 MAT (Carl Zeiss AG, Jena, Germany). The melt pool width and the measurement method are defined as shown in Figure 1b. Afterward, the samples are cut along the dotted lines in Figure 1a to determine the melt pool depth. To measure the melt pool depth, one-half of the sample is heat-treated at 1320 °C for 1 h in a vacuum furnace LHTM 250 (Carbolite Gero GmbH & Co. KG, Neuhausen, Germany). The heat treatment (HT) temperature was set based on preliminary experiments. The stripes are heat-treated in the $\alpha$-phase field, while the base plate with slightly less energy input and hence more aluminum content is heat-treated in the $\alpha + \gamma$-phase field. Consequently, the microstructure of the stripes is transformed into a fully lamellar (FL) microstructure, while the base plate remains nearly lamellar (NL), see Figure 3. More information on the HT of $\gamma$-TiAl parts built via PBF-EB can be found in [17]. Finally, the contrast of microstructures is used to determine the melt pool depth (Figure 3). To make the microstructures visible, the samples are ground and polished with a suspension consisting of 50 mL OPS, 50 mL distilled water, 10 mg KOH, and 10 mL 30% $H_2O_2$. The melt pool depth is measured at 12 positions 225 μm apart, which equates to three scan lines.

The geometry for evaluating the defects and aluminum evaporation Is shown in Figure 1e. Each stage of the square pyramid has a constant scan length between 45 mm and 2.5 mm, respectively, and is molten with the same parameter set as the other samples (see Table 2). Since the hatch direction is rotated by 90° for each layer, the evaluation cannot be divided into the x and y deflection of the electron beam. The samples for examining defects and aluminum evaporation are grinded and polished in the same way as described above. Light microscopy images for defect analysis are taken on a Zeiss AxioImager.M1m (Carl Zeiss AG, Jena, Germany). The area fraction of the gas porosity is evaluated by an automated script based on greyscale values and geometry. The aluminum content is measured via electron probe microanalysis (EPMA) on a Jeol JXA 8100 (JEOL, Akishima, Japan).

A semi-analytical heat conduction model is used for the numerical simulation of the melt pool as described in detail in [4]. The material properties of titanium aluminides are

summarized in Table 3. The shape of the electron beam is assumed to be Gaussian and its width is constant at 340 µm ($4\sigma$) or calculated according to Equation (5) [12].

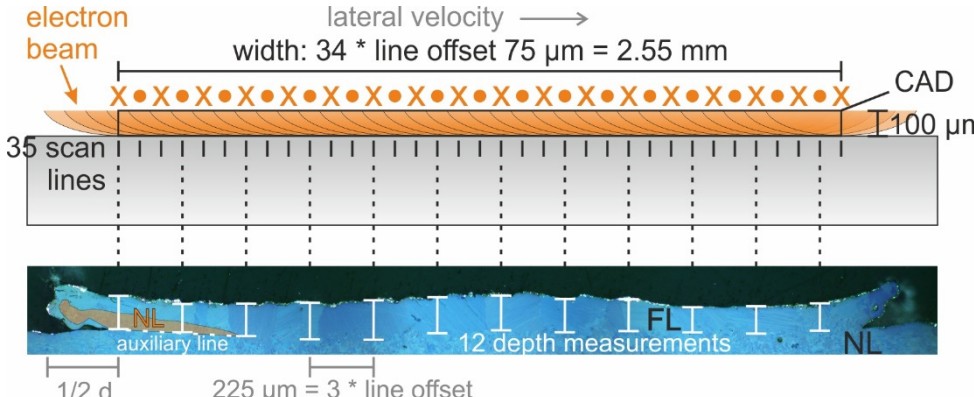

**Figure 3.** Schematic cut section of one of the stripes as depicted in Figure 1. The power-adapted beam size is displayed by the orange hemicycles (here for 900 W). The scan lines are separated by a line offset of 75 µm and follow a so-called "cross-snake" hatch pattern, indicated by the "x" and "●". Hence, the electron beam moves in and out of the presented plane. On the bottom, a cross-section of the sample is shown. The first scan line is determined by approximately half the width of the electron beam. Following the first scan line, the melt pool depth (fully lamellar microstructure, FL) is measured every 225 µm corresponding to every third scan line. Additionally, a small nearly lamellar (NL) area is found and measured at the bottom of the melt pool for the first few scan lines.

**Table 3.** Thermophysical properties for semi-analytic heat conduction model with TiAl.

| Property | Value |
|---|---|
| Thermal diffusivity [2] [$m^2$/s] | $6 \times 10^{-6}$ |
| Density [2] [$kg/m^3$] | 3533 |
| Specific heat [2] [J/(kg·K)] | 1160 |
| Absorption coefficient | 0.85 |
| Preheat temperature [K] | 1023 |
| Liquidus temperature [2] [K] | 1832 |
| Beam diameter ($4\sigma$) [µm] | 340 µm or Equation (5) |

[2] Values were determined by the LuFo InnoMat project 20T1712 founded by the Feder Ministry for Economic Affairs and Climate Action of Germany.

## 3. Results

### 3.1. Characterization of the Melt Pool

The melt pool during electron beam melting is characterized by the melt pool width, also called lateral extension, the melt pool depth, and the peak melt pool temperature. The focus of this study is on the melt pool width and depth. It is to note that it was not possible to determine either the width or the depth of the melt pool for 60 W beam power due to the small size of the molten area. The results of the light microscopic measurement of the melt pool characteristics are summarized in Figure 4. The average value of four samples (grey dots) are analyzed per beam power and a linear fit of these four measurements is presented (black dotted line).

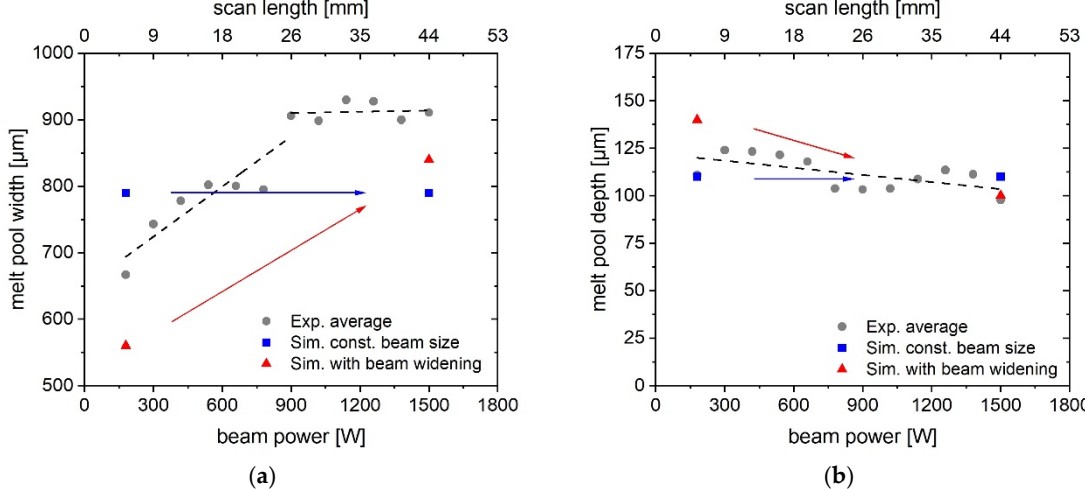

**Figure 4.** Melt pool width (**a**) and depth (**b**) over the beam power. For every beam power, the average of 4 data points (batch 1, batch 2, x-deflection, y-deflection) is plotted. The black dotted lines indicate a linear fit for the respective beam powers. The numerically calculated melt pool characteristics are shown for a constant beam diameter ($4\sigma$) of 340 μm (blue cubes & arrows) and with beam widening according to Equation (5) (red triangles & arrows).

The lateral extension steadily increases until 900 W. From there, the rise of the melt pool width is less pronounced up to the maximum tested beam power of 1500 W. The increased melt pool width is caused by the widening of the electron beam, which is discussed in detail in Section 4.2. The lateral extension of the melt pool is up to two times larger than the $4\sigma$ beam width. For example, a beam diameter of 402 μm leads to a width of 802 μm at 540 W beam power. Therefore, the melt pool can be characterized as persistent. The lateral extension scatters due to the stochastic powder bed, irregular melt pool borders (see Figure 1b), and manual measurement via light microscopy. The numerical calculation with a constant beam diameter ($4\sigma$) of 340 μm displays a constant melt pool width. Since the numerical calculations are based on a heat conduction model, geometry-induced changes in the heat conduction should be revealed in the simulations with constant beam diameter. In contrast, the numerical calculation with beam widening shows an increasing lateral extension with increasing beam power. However, the simulated lateral extension is smaller than the experimental values. Reasons for this difference include but are not limited to deviations of the beam size from the formulation by Equation (5), material properties taken from Ti-43.5Al-4Nb-1Mo alloy, and lack of latent heat in the semi-analytical heat conduction model. Nevertheless, the simulation reveals that an increasing electron beam diameter can lead to a wider melt pool, while the melt pool width is unaffected if the beam diameter is constant.

The average melt pool depth is determined at 12 measurement sites as shown in Figure 3. The average melt pool depth is mostly in the range of 100–125 μm. This result is expected because the chosen parameter is towards the low energy end of the process window to avoid extensive aluminum evaporation and hence should be just sufficient to melt slightly more than one layer of thickness. The scattering of the melt pool depth can be attributed to small fluctuations during the PBF-EB process and the fact that only a single cut plane is examined. The numerical calculations of the melt pool depth fit well with the experimental values. The beam widening shows only a slight change in the melt pool depth. Therefore, the main influence of the increasing beam diameter is on the lateral extension of the melt pool.

### 3.2. Microstructure and Defects

#### 3.2.1. Starting Area

Generally, the heat-treated melt pool samples display a homogenous, fully lamellar top layer. However, at the beginning of each stripe, within the first few scan lines, there is a nearly lamellar area at the bottom of the melt pool (see Figure 3), indicating reduced aluminum evaporation. During these first few scan lines, the heat transfer from previous scan lines increases until a steady state is reached. The peak temperature at the beginning of each stripe is reduced and hence less aluminum evaporates. The nearly lamellar area is measured for each strip and the results are shown in Figure 5. The nearly lamellar area is independent of the deflection direction and powder batch. It declines up to 900 W and rises for higher beam power.

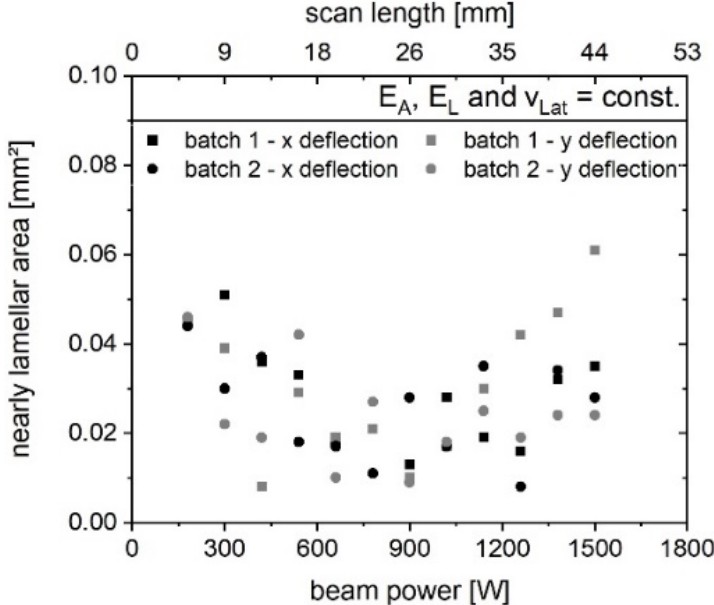

**Figure 5.** The nearly lamellar area at the bottom of the melt pool during the beginning of the melting over the beam power for both powder batches in x and y deflection. The NL area decreases with rising beam power until around 15 mA and increases for higher beam power from this point on.

#### 3.2.2. Porosity and Aluminum Content

Figure 6 displays the gas porosity of the bulk samples (see Figure 1e) in dependence on the beam power on the left and the aluminum content on the right. The powder properties are indicated by the dotted lines. Misconnections are not found in the samples. The gas porosity declines from roughly 0.5% for small beam powers to 0.1% around 900 W. For higher beam powers, it increases to 0.3–0.4%. The aluminum content shows a similar trend. After an initial drop up to 900 W, it rises again for higher beam powers. The slight deviations between batches 1 and 2 can be attributed to the different aluminum content in the powder. Both microstructural features indicate that the melt pool becomes hotter, larger, and longer-lasting up to a peak at approximately 900 W followed by a reduction of these melt pool characteristics for higher beam powers. These results fit well with the previous findings of the nearly lamellar starting area. The connection between the melt pool dimensions and the final microstructure features is discussed in the following section.

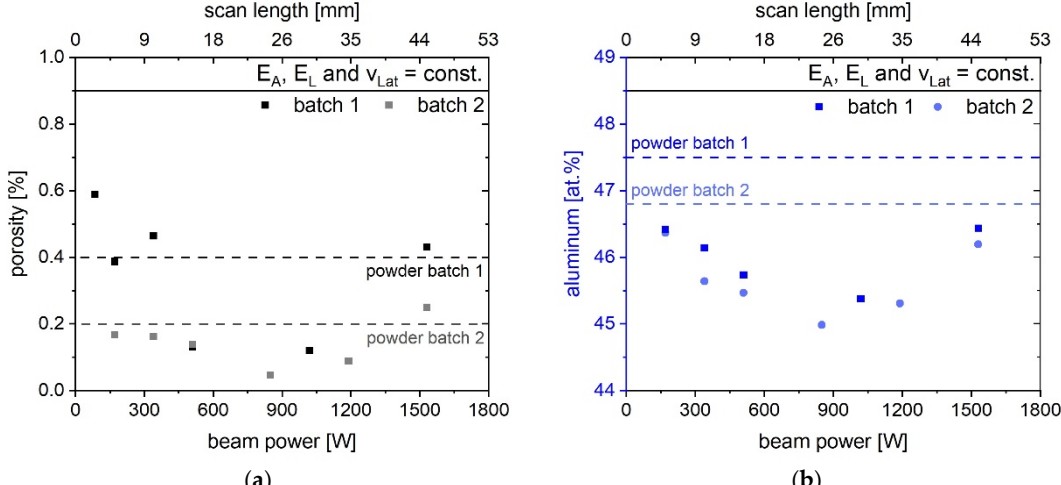

**Figure 6.** Porosity (**a**) and aluminum content (**b**) over the beam power for both powder batches. The porosity and the aluminum content decrease with increasing beam power up to 900 W. Subsequently, both values rise for higher beam power. The dotted lines indicate the powder properties: the gas porosity inside the powder (**a**) and the measured aluminum content (**b**) of the powder.

## 4. Discussion

### 4.1. Correlation between Melt Pool and Microstructure

In Figure 7, the experimental results from the previous sections are summarized in the form of average values to display the principal relationships between beam power, melt pool, and resulting microstructure. Area energy, line energy, beam return time, and lateral velocity are constant to isolate the effect of the beam power. To discuss these results, it is important to bear in mind that aluminum evaporation mainly depends on the peak temperature of the melt pool [12], the area of the melt pool [13], and the time in the liquid state [13], also called melt pool lifetime. It should be noted that these melt pool properties are not independent of each other. For instance, a hotter melt pool is also larger and stays longer in the liquid state. The melt pool lifetime also determines the time for entrapped gas, introduced by the powder or process, to rise and escape from the melt pool and hence the final porosity. Lastly, the nearly lamellar area is closely linked to the aluminum evaporation as well since the overall Al loss influences the Al-enriched zone of the first few melt lines.

The microstructural properties from the experiments can be divided into two regions. From 180 to 900 W, the aluminum evaporation increases, while the nearly lamellar area and the gas porosity decrease with increasing beam powers. This indicates, that the peak melt pool temperature gets higher and the melt pool becomes broader and longer-lasting. These findings fit well with the measured and calculated lateral melt pool extension, which is also increasing. After a minimum around 900 W, the gas porosity and nearly lamellar area steadily rise, whereas the aluminum evaporation declines. At the same time, the lateral melt pool extension is still slightly increasing. Consequently, the melt pool has to have a lower peak temperature and stay shorter in the liquid state to account for the reduced aluminum evaporation and increased gas porosity. To explain these opposing trends of the evolution of the melt pool and the resulting microstructure, a closer assessment of the beam diameter and energy input via the electron beam is conducted in the following section.

Contrary to the other results discussed, the melt pool depth does not show a strong impact on the beam power. Therefore, the melt pool depth is less sensitive to the beam diameter than all other features. The penetration depth for 60 kV electrons in titanium is approximately 10 μm [18]. Therefore, most of the melt pool depth is molten via heat conduction [19], and the influence of the broader electron beam for increasing beam powers is not as crucial as for the lateral melt pool extension or the aluminum evaporation.

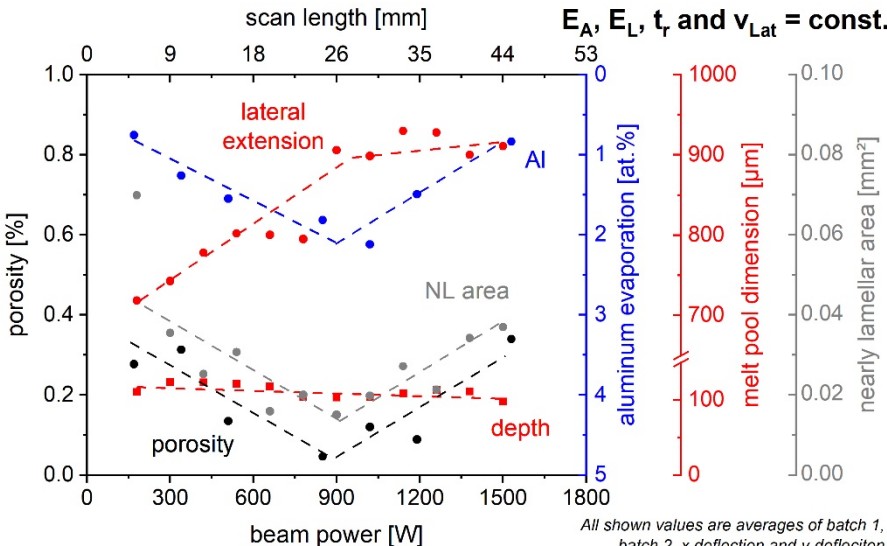

**Figure 7.** The correlation between melt pool depth, lateral extension, gas porosity, aluminum evaporation, and the nearly lamellar area is shown. Average experimental values are given by the symbols, while the dotted lines represent the overall trend. The area energy $E_A$, line energy $E_L$, return time $t_r$, and lateral velocity $v_{Lat}$ are constant. For small beam powers up to 900 W, the gas porosity (black) and nearly lamellar area (grey) decrease, whereas the lateral extension (red) and aluminum evaporation (blue) increase. The gas porosity and nearly lamellar area rise for higher beam powers, while the Al evaporation is reduced. At the same time, the lateral extension of the melt pool is constant.

### 4.2. Beam Diameter and Energy Input

For the theoretical calculations concerning the beam diameter and energy input, the beam is assumed to be perfectly round and its diameter is calculated according to Equation (5). The values presented here are for this specific case, an A2X from GE Additive. However, it is strongly recommended to consider the beam diameter as an important scanning parameter for all PBF-EB machines. The beam power density $q_A$ is described in Equation (6). The absorption coefficient $\eta$ is set to 0.85 [4,12].

$$q_A = \frac{\eta * P}{\pi * \left(\frac{d}{2}\right)^2} \tag{6}$$

The dependence of the beam power density on the beam power is shown in Figure 8. Additionally, a cut in the *x-z*-plane ($y = 0$ and $y_0 = 0$) for a resting beam ($x_0 = 0$) of the Gaussian beam power distribution $q_A(x,y)$ within the electron beam is plotted according to Equations (7) and (8) [3,4,18]. The variance $\sigma$ is $\frac{1}{4} d$.

$$q_A(x,y) = \eta * I(x,y) * P_B \tag{7}$$

$$I(x,y) = \frac{1}{2\pi\sigma^2} \exp\left(\frac{(x - x_0)^2 + (y - y_0)^2}{2\sigma^2}\right) \tag{8}$$

Figure 8 displays a rise in the beam power density, which is a measure of the energy density of the electron beam from 180–780 W. Simultaneously, the electron beam widens and the energy density in the center of the beam rises (see Gaussian beam power distribution in Figure 8). This leads to higher peak temperatures and a broader melt pool, enhancing aluminum evaporation and reducing gas porosity. This agrees well with the experimental findings.

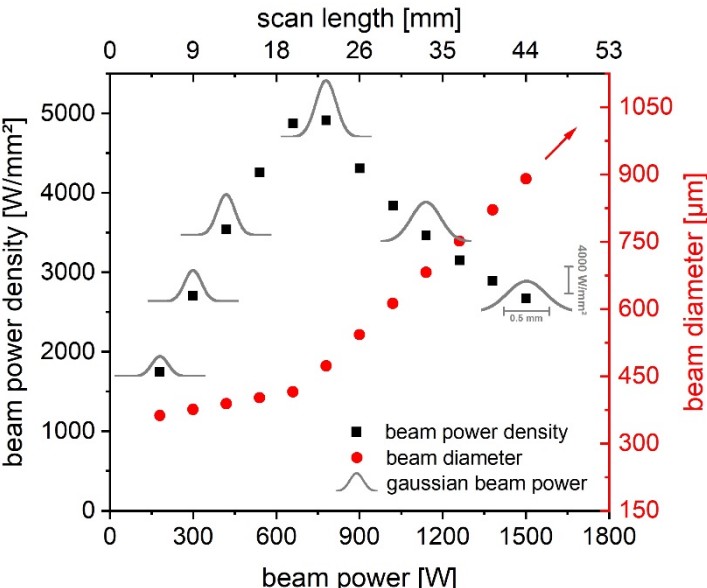

**Figure 8.** Calculated beam power density (Equation (6), black squares), beam diameter (Equation (5), red dots), and Gaussian beam power distribution (Equation (7), grey lines) plotted over the beam power.

For beam powers larger than 780 W, the beam power density decreases. At the same time, the beam widening is enhanced and the energy density in the center of the electron beam is reduced. The lower peak energy input leads to less overheating during the initial melting and therefore reduces the peak temperatures. Consequently, the aluminum evaporation decreases and the melt pool solidifies faster, which results in more gas porosity. Meanwhile, the broadening electron beam increases the interaction area with the powder bed and melt pool, resulting in a more horizontal distribution of the energy and finally leading to a wider melt pool. In conclusion, the melt pool possesses lower peak temperatures and is shorter-lived, leading to less aluminum evaporation and more gas porosity, while the lateral extension rises due to the wider spread energy input.

The simultaneous development of the beam power density and peak power (Figure 8, black and grey) to the aluminum evaporation, gas porosity, and nearly lamellar area (Figure 7, blue, black, and grey) is striking. Moreover, the development of the lateral melt pool extension (Figure 7, red) can be connected to the increase in the beam diameter (Figure 8, red). Overall, these results show a strong correlation between the energy input via the electron beam and the resulting melt pool and microstructure.

## 5. Conclusions

The beam diameter has a strong influence on the PBF-EB process. Despite keeping line energy, area energy, return time, or lateral velocity constant, changes in melt pool characteristics and microstructure are observed. The gas porosity varies between 0.5 and 0.1% and the aluminum content is in a range of 0.8–2.1%. These changes are attributed to the power dependence of the beam diameter, which is currently not considered when calculating line energy or area energy. The widening of the electron beam with increasing beam power is affecting both, the power distribution of the beam as well as the resultant peak temperature in the powder bed. In consequence, the varying melting conditions have a strong impact on melt pool geometry and microstructure.

Knowledge about the beam diameter and its dependence on the beam power is mandatory to control the PBF-EB process in detail. Quantifying the size and the shape as well as the power distribution of the electron beam is necessary for the scaling of process parameters from simply shaped geometries with uniform scan lengths to complex parts with various scan lengths. In addition, the determination of the system-dependent beam

diameter is highly recommended when transferring process parameters between different PBF-EB machines to provide almost equivalent process conditions necessary for on-demand microstructure and properties.

**Author Contributions:** Conceptualization, M.R. and M.F.; methodology, M.R.; software, C.B.; validation, M.R. and C.B.; formal analysis, M.R. and C.B.; investigation, M.R. and C.B.; resources, M.F.; data curation, M.R. and C.B.; writing—original draft preparation, M.R.; writing—review and editing, M.F., C.B. and C.K.; visualization, M.R. and M.F.; supervision, M.F. and C.K.; project administration, M.F.; funding acquisition, M.F. All authors have read and agreed to the published version of the manuscript.

**Funding:** The Chair of Materials Science and Engineering for Metals gratefully acknowledges the German Research Foundation—Project ID 61375930—SFB 814—"Additive Manufacturing" TP B02 for financial support.

**Institutional Review Board Statement:** Not applicable.

**Informed Consent Statement:** Not applicable.

**Data Availability Statement:** The data presented in this study are available on request from the corresponding author.

**Acknowledgments:** We are grateful to Sabine Michel from the Chair of Materials Science and Engineering for Metals (FAU, Erlangen, Germany) for conducting the EPMA measurements and Maria Schroeder from Neue Materialien Fuerth (Fuerth, Germany) for conducting the heat treatments.

**Conflicts of Interest:** The authors declare no conflict of interest.

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
