# Peer review of "Impact of the Power-Dependent Beam Diameter during Electron Beam Additive Manufacturing: A Case Study with γ-TiAl"

_applsci, doi:10.3390/app122111300_

Round 1
Reviewer 1 Report
The authors investigated the impact of the power-dependent beam diameter during electron beam additive manufacturing. It is a good paper and should be published after minor revision.
1. “The lateral extension steadily increases until 900 W. From there, the rise of the melt pool width is less pronounced up to the maximum tested beam power of 1500 W.” please, better explain this reason.
2. It is recommended that all tables be changed to three-line tables.
3. In section 3.2, please add the title number before “Starting area” and “Porosity and aluminum content”.
4. Several grammar errors exist. Please revise carefully.
Author Response
Point 1: The lateral extension of the melt pool is closely linked to the energy input during melting. The citation you referring to is from the results section, where I simply describe the results found. The connection between energy input and melt pool is discussed in section 4.2. To make it easier for the reader to connect the parts I added the sentence “The increased melt pool width is caused by the widening of the electron beam, which is discussed in detail in section 4.2.” after the citation you mentioned.
Point 2: All tables are changed accordingly.
Point 3: Title numbers are added.
Point 4: Thank you for the remark. We revised the manuscript carefully.
Reviewer 2 Report
Authors have carried out good amount of work, in my view manuscript contains good quality work and can be accepted for publication in its present form.
Author Response
Thank you for the kind feedback.
Reviewer 3 Report
This study aims to demonstrate typical scaling issues when applying process parameters to scan lengths up to 45 mm using the example of a 4th generation γ-TiAl alloy. During the additive 17 manufacturing process line energy, area energy, return time, and lateral velocity are kept constant 18 by adjusting beam power and beam velocity to various scan lengths. The experiments reveal undesired changes in melt pool geometry, gas porosity, and aluminum evaporation by increasing the beam power. It can be accepted after revision.
1. It is suggested to add references in recent three years to highlight the innovative of articles.
2. The conclusion part needs to be condensed to reflect some quantitative conclusions.
3. The introduction part needs to be refined to reflect innovation.
Author Response
Point 1: I updated one of the review papers from 2016 with a more recent one from 2022. Now 7/19 references are from the recent three years (2020 – 2022) and 14/19 references are from the recent six years (2017 – 2022). Also, it is to note, that there is not a lot of research about the beam diameter in electron-based powder bed fusion.
Point 2: The changes in microstructure depend strongly on the used system and material. Therefore, it is difficult to give precise values on the change of microstructure for a defined amount of beam widening. That is why we focused more on the general consequences for the PBF-EB process. Nevertheless, I included some quantitive results in the conclusion “The gas porosity varies between 0.5 % and 0.1 %, and the aluminum content is in a range of 0.8 at.% to 2.1 at.%.”.
Point 3: We tried to point out more clearly what is innovative about our research. Espescially the last paragraph of the introduction is refined, which shows the aim of our study. Therefore, we added some phrases like “Hence, this novel approach […]” and sentences like “For the first time, the connection between beam diameter, power density, melt pool, and microstructure is systematically analyzed for EB-PBF.”